psychology

imitation, mimicry, alignment, social coordination

**Author for correspondence:**
Jarosław R. Lelonkiewicz
e-mail: jlelonki@gmail.com

# Does it pay to imitate? No evidence for social gains from lexical imitation

Jarosław R. Lelonkiewicz[1], Martin J. Pickering[2] and Holly P. Branigan[2]

[1]Cognitive Neuroscience Area, Scuola Internazionale Superiore di Studi Avanzati, via Bonomea 265, Trieste 34136, Italy
[2]School of Philosophy, Psychology and Language Sciences, University of Edinburgh, 7 George Square, Edinburgh, EH8 9JZ, UK

 JRL, 0000-0002-3802-3393

According to an influential hypothesis, people imitate motor movements to foster social interactions. Could imitation of language serve a similar function? We investigated this question in two pre-registered experiments. In Experiment 1, participants were asked to alternate naming pictures and matching pictures to a name provided by a partner. Crucially, and unknown to participants, the partner was in fact a computer program which in one group produced the same names as previously used by the participant, and in the other group consistently produced different names. We found no difference in how the two groups evaluated the partner or the interaction and no difference in their willingness to cooperate with the partner. In Experiment 2, we made the task more similar to natural interactions by adding a stage in which a participant and the partner introduced themselves to each other and included a measure of the participant's autistic traits. Once again, we found no effects of being imitated. We discuss how these null results may inform imitation research.

## 1. Introduction

People are prolific imitators—we copy the behaviour we see in others, from gestures to facial expressions to different aspects of language [1,2]. Thus far, much research has focused on motor mimicry—the spontaneous copying of movements such as foot shaking or face touching. There is evidence that this form of imitation may be used to improve our interactions with others. In a landmark study by Chartrand & Bargh [3], participants described photographs together with a partner. Throughout the task, the partner—in fact a secret confederate of the experimenters—either copied the posture, movements and mannerisms of the participant

(imitation group) or maintained a neutral body position and did not copy (control group). Participants then filled out a questionnaire evaluating the interaction and the partner. Analysis revealed that the group that was imitated rated the interaction as smoother and the partner as more likeable compared to the control group. Several other studies have reported similar findings, thereby linking motor mimicry to greater empathy, feelings of closeness and an increased tendency to help and cooperate (for reviews, see [1,4]).

It has been long hypothesized that the imitation of language could play a similar role. Specifically, several authors have suggested that speakers engage in such imitation to satisfy the need of maintaining positive social relations. In other words, the mechanism that underlies linguistic mimicry has been hypothesized to be, at least in part, of a social nature ([5–7]; see also [8,9]). Importantly, this idea rests on a critical assumption—that imitating language brings about certain social gains and indeed can improve interactions with others. Yet, in this regard, the current literature paints a complicated and somewhat confusing picture.

To begin with, most of the existing evidence has been obtained through correlational studies. For example, speakers who happen to use similar language also happen to more often cooperate with each other [10] and form romantic relationships [11]. Likewise, phonetic convergence is associated with greater self-reported closeness toward the partner [12]. However, because they are correlational, such findings do not demonstrate a causal role of imitation in achieving social gains.

In another line of research, studies investigated whether speakers engage in more linguistic mimicry when the social interaction needs to be repaired. Balcetis & Dale [5] found that participants interacting with an annoyed partner exhibited higher levels of syntactic mimicry (i.e. repetition of the sentence structure used by the partner) than those who interacted with a patient partner. More recently, Hopkins & Branigan [13] showed that children who had experienced ostracism more often copied the lexical choices of their partner than children who had not experienced ostracism. Although such evidence suggests that linguistic mimicry might be used with the goal of achieving social ends, it does not prove that this form of imitation can actually support social interactions or benefit the imitator otherwise.

Only a handful of studies have directly tested the social consequences of mimicking language. Unfortunately, their data do not provide a clear picture. First, the interpretation of some key experimental findings is complicated by methodological shortcomings. For example, a highly cited paper by van Baaren *et al.* [14] reported that waitresses received higher tips when they repeated verbatim their customers' orders, as compared to when they did not repeat them. However, it is unclear whether the behaviour of waitresses across experimental conditions was controlled (the paper mentions that waitresses were asked to behave consistently across conditions, but does not report any checks of whether these instructions were followed). Thus, we cannot be sure if the observed differences in tipping were truly due to the imitation manipulation, or some other factors (e.g. waitresses could have used fewer words in the non-repetition than in the repetition condition, potentially breaching politeness norms). A similar critique applies to Müller *et al.* [15], who found that a confederate was more likely to be helped by strangers after copying both their motor behaviour and language.

Another complication is that there are inconsistencies between experimental findings reported in the literature. In Abrahams *et al.* [16], participants engaged in a picture description task interacting either with a confederate who copied their choice of grammatical structures or with a confederate who did not do so. Being syntactically imitated led to an increase in cooperativeness, as indicated by the fact that the imitated participants spent more time helping the confederate with an extra task (although this was the case only for participants who were native speakers of the language used in the study, and not for non-native speakers). However, an earlier study from the same laboratory found no such effect despite using the same manipulation [17]. To further complicate the picture, in the first experiment reported by Schoot *et al.* [18], syntactic mimicry led to decreased liking, but in a second experiment using the same paradigm it had no reliable effect.

Our previous work revealed a similarly complex pattern of results with regard to lexical imitation [2]. In two laboratory experiments, we asked participants to take turns naming pictures and matching pictures to a name provided by a partner. Participants were seated in different rooms and carried out the task using a computer. On certain trials, the partner named a picture that had previously been named by the participant, and we manipulated whether the partner used the same word for the picture (imitation group; e.g. participant says 'stereo', partner says 'stereo') or a different word (counter-imitation group; e.g. participant says 'stereo', partner says 'boombox'). Unknown to participants, the partner's responses on those trials were in fact generated by a computer. Next, participants completed a questionnaire evaluating the partner and the interaction and played a short

investment game measuring their willingness to cooperate. Although we did observe some positive effects of lexical imitation, our findings did not replicate between experiments: Whereas in Experiment 1, imitated participants rated the interaction more positively than counter-imitated participants, in Experiment 2, there was no such difference; instead, in Experiment 2, lexical imitation led to an increase in cooperative behaviour—a tendency that was absent in Experiment 1.

In sum, it is currently difficult to understand what role linguistic imitation plays in social functioning. This is because only a small number of experiments have directly tested the consequences of imitating language, and because these studies yielded an inconsistent pattern of results. We argue that to resolve this fundamental question, it is imperative to collect further evidence, particularly by running rigorous experimental studies aiming to replicate the reported effects.

In this paper, we report two pre-registered online experiments attempting to replicate the finding that lexical imitation encourages a more positive evaluation of the interaction ([2]; Experiment 1). With the exception of being carried out online, Experiment 1 closely followed the design used in our previous work: participants were asked to interact with a partner and complete the picture naming/matching task; the partner, in fact a computer program, either imitated or counter-imitated their lexical choices. We measured whether this manipulation affected how participants evaluated the interaction, how they evaluated the partner, and to what extent they cooperated in an investment game. Experiment 2 was identical to Experiment 1, except that we added a measure of the participant's autistic traits and made the task more akin to natural interactions by adding an initial stage in which participants and partners introduced themselves to each other. In Experiments 1 and 2, we investigated whether lexical imitation leads to a more favourable evaluation of the interaction or the partner, and whether it increases cooperativeness. In addition, in Experiment 2, we tested if any social effects of imitation would be modulated by the prevalence of autistic traits in the individual.

# 2. Experiment 1

## 2.1. Methods

The experimental design, hypotheses, sample size calculations, data collection and analysis plan were pre-registered on Open Science Framework (OSF). See the project's OSF webpage for the pre-registration files, as well as experimental materials, data and analysis scripts (https://osf.io/z34m8/).

## 2.2. Participants

We recruited 167 participants through the online recruitment platform Prolific (https://www.prolific.co/). Participants were native speakers of British English, born and currently living in the UK, aged 18–35. Prior to analysis, we excluded 12 participants who produced corrupt/nonsensical data (1 participant) or guessed some of the hypotheses of the study (11 participants), resulting in the final sample of 155 participants (80 in the imitation, 75 in the counter-imitation condition).

This exceeded the sample suggested by our pre-registered calculations: we assumed a medium effect size $d = 0.50$, alpha = 0.017, 80% power, one-tailed between-group $t$-test comparison and estimated the desired sample size at 144 participants (note that we were conservative in choosing the effect size value: In Lelonkiewicz [2], Experiment 1, the increase in the evaluation of social interaction observed in the imitation group was associated with an effect size of $d = 0.61$; further, the choice of alpha = 0.05/3 = 0.017 accommodated the fact that we investigated the effect of imitation for three different measures; calculations were carried out using GPower v. 3.1; [19].

## 2.3. Design, materials and procedure

The study consisted of (1) the picture naming/matching task, (2) investment game, (3) partner and interaction evaluation questionnaire and (4) debriefing questions. The study was advertised on Prolific. It was visible only during specific times and only to individuals who met the sample characteristics (as specified in the Participants section). Participants viewed the description of the experiment (which also served as an informed consent form) and chose whether to take part. They were promised £1.50 for completing the study with the possibility of winning a bonus of up to £1

depending on performance in the investment game (in fact, they all were paid £2.50). At this stage, participants were given the cover story: the experimental description stated that they would work together with another participant, and that the task was accessible only during certain times to increase the chance that multiple users were available. In addition, on starting the experiment, they saw a sequence of screens reinforcing this story ('Searching for available clients', onscreen for 8000 ms; 'There are 5 clients connected to the server' for 2000 ms; 'Connecting to the other client. Please wait' for 1500 ms; 'Player1: [participant Prolific ID] Player2: [randomly generated partner's ID]' for 4000 ms). The study took 10–20 min.

### 2.3.1. Picture naming/matching task

Next, participants read the instructions for the picture naming/matching task. These explained that participants would take turns naming pictures and matching pictures to a name, and that they would perform the task together with another person over a network connection. In fact, there was no connection and participants interacted with a preprogrammed computer script.

The task involved 20 experimental trials in which a target picture appeared alongside a distractor and 20 filler trials consisting of two distractor pictures. Target pictures had two similarly frequent names (e.g. for the eraser/rubber picture, 'eraser' was chosen by 42% and 'rubber' by 58% of responders), and distractors had a single dominant name (greater than 92%). Each target picture appeared twice and each distractor appeared once. In total, we used 10 unique targets and 60 unique distractors (selected from the Bank of Standardized Stimuli; [20]). Name frequencies for targets were calculated based on a pre-test in which 68 further native British English speakers named 42 pictures by writing a single word below each picture; distractor name frequencies were based on the norms from Brodeur *et al.* [20]. See the project's OSF page for stimuli lists and full information on name frequencies.

On each trial, two pictures appeared on the screen. If it was participants' turn to name, they saw a message asking them to name either the left or right picture, and they typed the name for this picture into a text box below (they were instructed to use a single word and avoid spelling mistakes; the message and pictures remained onscreen until participants submitted the response). Next, they were asked to wait for the partner's response, implying that their partner was about to match a picture to the name they had just provided. The picture named by the participants then became highlighted, followed by a feedback message suggesting that the partner's matching response was correct. To make it appear that participants interacted with a human partner, we randomly varied the delay after which the picture was highlighted (3000–4000 ms after participants submitted their response; the feedback message appeared after a further 1700 ms and remained onscreen for 2500 ms). If on a given trial, it was the participants' turn to match, a name for one of the pictures—allegedly produced by the partner—was displayed below the pictures (the name appeared 5000–6000 ms after picture onset). Participants were asked to select the picture that matched the name, using the arrow keys. The selected picture was then highlighted (immediately after the response) and a feedback message informed participants whether their response was correct (timing as on naming trials).

Trials were organized in a turn sequence where there were two fillers between each experimental trial (experimental naming turn → filler matching turn → filler naming turn → experimental matching turn, and so on). The experimental trials within a sequence shared the same target picture. For example, if participants named the eraser/rubber picture on the experimental naming turn, they would then match this picture on the experimental matching turn.

Crucially, participants were randomly assigned to either the imitation or counter-imitation condition. In the imitation condition, the computer recorded the name entered by the participant for the target picture on the experimental naming turn (e.g. 'eraser'). Then, when the target picture reappeared on the experimental matching turn, the computer displayed this name (eraser). It therefore seemed as though the partner was responding with the same name.

In the counter-imitation condition, the computer recorded the name entered by the participant and compared it with the two words that had been identified in the pre-test as alternative names for this picture ('eraser' versus 'rubber'). The computer first checked if the name entered by the participant matched the word that was the more frequent alternative in the pre-test (eraser). If so, the computer displayed the other name as the partner's response (rubber). If not, then the computer displayed the more frequent name. It therefore seemed as though the partner was

responding with a different (though appropriate) name than the name previously used by the participant.[1]

### 2.3.2. Investment game

After the picture naming/matching game, participants read the instructions for the investment game. They were told they would play the game together with the human partner, but in fact, they again completed the task alone. The game consisted of six turns. On each turn, each player was given 40 credits (1 credit = 0.1p) that they could invest into a shared pool, or keep to themselves. The amount in the pool was multiplied by a factor and then divided between the two players at the end of the turn. The multiplication factor varied between turns: for turn one, we used a factor of 1.9; the factors for the remaining five turns were 1.5, 1.7, 1.9, 2.1 and 2.3, presented in a randomized order.[2] The payoff from each turn consisted of the earnings from the pool plus the credits the participant decided to keep. Participants were not informed about the outcome of a given turn; after deciding how many credits to invest or keep, they proceeded to the next turn. However, they were told that one turn would be picked at random and that the outcome from that turn would be used to calculate the monetary reward for the investment game.[3] The investment game was self-timed (i.e. participants proceeded to the next screen after a button press/response). The investment game and the picture naming/matching task were implemented using a custom-made experimental script (based on jsPsych v. 5.0.3; [22]) and hosted on an Apache server.

### 2.3.3. Post-interaction questionnaires

Next, participants completed two questionnaires measuring their evaluation of the interaction and the partner. For the first questionnaire, we used the original tool from Chartrand & Bargh [3]. Participants answered two items asking about the smoothness of the interaction and the likability of the partner ('How smoothly would you say your interaction went with the other participant?'; 'How likable was the other participant?'). The remaining eight items were fillers asking about various aspects of the experiment (e.g. 'How familiar were the entities in the pictures?'). Items were measured on a 1–9 scale with polar adjectives as anchors for the low and high ends (e.g. 'extremely rare' versus 'extremely familiar'). The second questionnaire involved six items evaluating the interaction, intermixed with a further 11 items concerning the partner (taken from the Reysen Likability Scale; [23]). Items were measured on a 1–9 Likert scale with anchors ranging from 'very strongly disagree' to 'very strongly agree'. Across the two questionnaires, in total seven items evaluated the interaction and 12 items evaluated the partner.

### 2.3.4. Debriefing questions

In the end, participants answered three open-ended questions probing whether they guessed the aims of the study ('What do you think was the purpose of the experiment?'; 'What do you think of the partner with whom you interacted?'; 'What did your partner do during the naming/matching game?'). They were then debriefed about the aims of the study, thanked and paid. The post-interaction questionnaires and debriefing questions were implemented with Jisc/Online surveys (https://www. jisc.ac.uk/).

---

[1]Although the experiment was to an extent resistant to mistakes (i.e. it ignored any non-letter characters added to a response and was indifferent to lower/upper case; participants were instructed to avoid spelling mistakes), there was a possibility that an unusual spelling or spelling mistake could interfere with our imitation/counter-imitation manipulation. Specifically, in the imitation condition, responses were copied verbatim, thus potentially including minor spelling mistakes (e.g. participant typed 'casette' and the experiment showed 'casette' on the partner's naming turn); in the counter-imitation condition, such mistakes could impair the ability of the experiment to recognize the word used by the participant, leading it to show the same, but correctly spelt word, rather than an alternative name for the picture (e.g. participant typed 'casette' and the program showed 'cassette', rather than 'tape'). Importantly, however, spelling mistakes occurred at a very low rate (1.29% of all experimental trials in Experiment 1 and 1.35% in Experiment 2 involved unusual spelling or a spelling mistake), and so it is unlikely that they interfered with our manipulation to a noticeable extent.

[2]We used a fixed factor on turn one but randomized the order of factors on turns two to six so as to replicate the parameters of the game used in Lelonkiewicz [20].

[3]We kept the participants blind to the outcome of each individual turn to increase the chance that their decisions would reflect the tendency to cooperate induced by imitation or counter-imitation in the picture-naming/matching task. Blind set-ups have been shown to yield similar results to standard decision-making games (e.g. [21]). Once again, this parameter was as in Lelonkiewicz [20].

**Table 1.** Experiments 1 and 2: interaction, partner and cooperativeness scores across the counter-imitation and imitation condition. The condition means reported with 95% CI (values in square brackets).

| | | partner | interaction | cooperativeness |
|---|---|---|---|---|
| Experiment 1 | counter-imitation | 5.57 [±0.28] | 6.27 [±0.32] | 25.48 [±2.46] |
| | imitation | 5.60 [±0.28] | 6.15 [±0.32] | 25.94 [±2.18] |
| Experiment 2 | counter-imitation | 6.03 [±0.27] | 6.56 [±0.31] | 26.89 [±2.32] |
| | imitation | 6.10 [±0.29] | 6.75 [±0.29] | 25.79 [±2.38] |

## 2.4. Results

We asked whether being lexically imitated encourages a more positive evaluation of the interaction and/ or the partner. To investigate this, we analysed the data from the post-interaction questionnaires. For each participant, we calculated a single-interaction evaluation score and a single-partner evaluation score by averaging responses on the relevant items. We then compared these scores between groups and found that participants who were imitated did not differ from those who were counter-imitated in terms of their evaluation of the interaction ($U = 3165.5$, $p = 0.554$; all tests reported in this section are Mann–Whitney $U$ tests, one-tailed) or their evaluation of the partner ($U = 2985.5$, $p = 0.960$). See means in table 1 and score distributions in figure 1.

In a similar vein, we tested if lexical imitation increases the tendency to cooperate. For each participant, we calculated a cooperativeness score by averaging the contributions this participant made to the shared pool of credits across all turns of the investment game. We found that the imitated and counter-imitated participants did not differ in terms of their cooperative tendencies ($U = 2951$, $p = 0.862$; table 1 and figure 1).

## 2.5. Discussion

Experiment 1 failed to find any social effects of imitation: participants who were lexically imitated did not differ from those who were counter-imitated in terms of their evaluation of the interaction or the partner and did not differ in terms of cooperativeness. At this stage, we speculated that our online paradigm might not have provided enough social context: participants interacted with a partner who was entirely anonymous, which could have made it difficult for them to process imitation as a social cue (we elaborate on this possibility in the General Discussion). Thus, in Experiment 2, we modified the paradigm to add an initial stage where participants introduced themselves to the partner (and vice versa), thus mirroring natural social interactions in which strangers typically begin by exchanging greetings and sharing some minimal information about each other.

In addition, we asked if the hypothesized effects of imitation could be occluded by certain characteristics of the individual. Previous research suggested that the relationship between social factors and lexical imitation is modulated by the prevalence of autistic traits. Specifically, children who are generally less attuned to social cues also show less sensitivity to linguistic mimicry as an affiliative behaviour [24]. We wanted to limit the potential impact of such differences on our ability to detect imitation effects. To this end, in Experiment 2, we added a further questionnaire measuring the prevalence of autistic traits in the individual (AQ10: Autism-Spectrum Quotient; [25]) and controlled for the possible modulatory effect of this measure in our analyses.

# 3. Experiment 2

## 3.1. Methods

See the project's OSF webpage for the pre-registration files, experimental materials, data and analysis scripts (https://osf.io/z34m8/).

## 3.2. Participants

We recruited a further 166 participants from the same population and on the same basis as in Experiment 1. We excluded five participants who produced corrupt/nonsensical data and 14 who guessed the

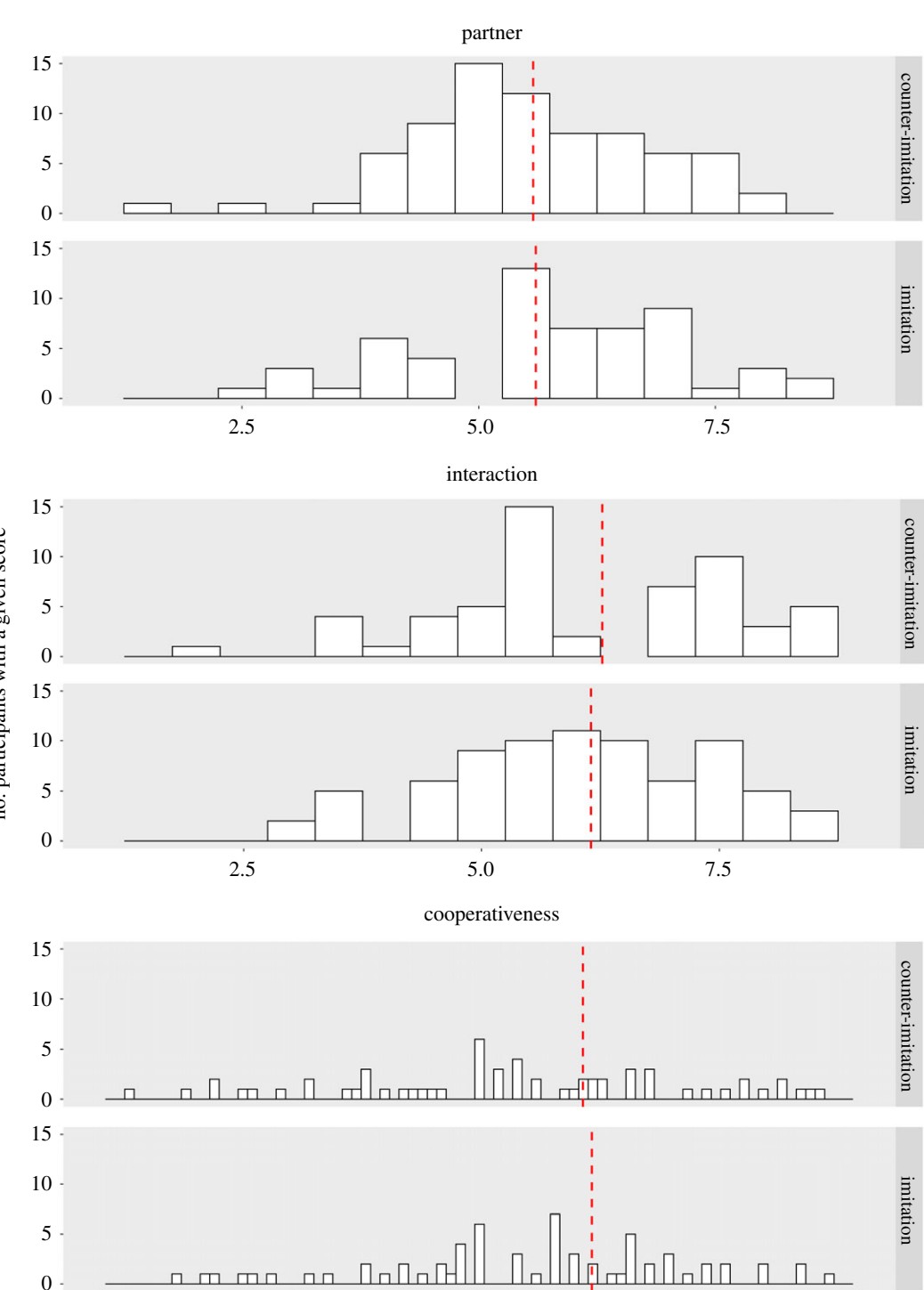

**Figure 1.** Experiments 1 and 2: histograms illustrating the distribution of interaction, partner and cooperativeness scores across the imitation and counter-imitation condition.

hypotheses of the study, leaving 147 participants in the analysis (75 imitation, 72 counter-imitation condition).

### 3.3. Design, materials and procedure

Experiment 2 replicated Experiment 1 with two exceptions: first, immediately before the picture naming/matching game, we added a message asking participants to type in a self-introduction where they briefly

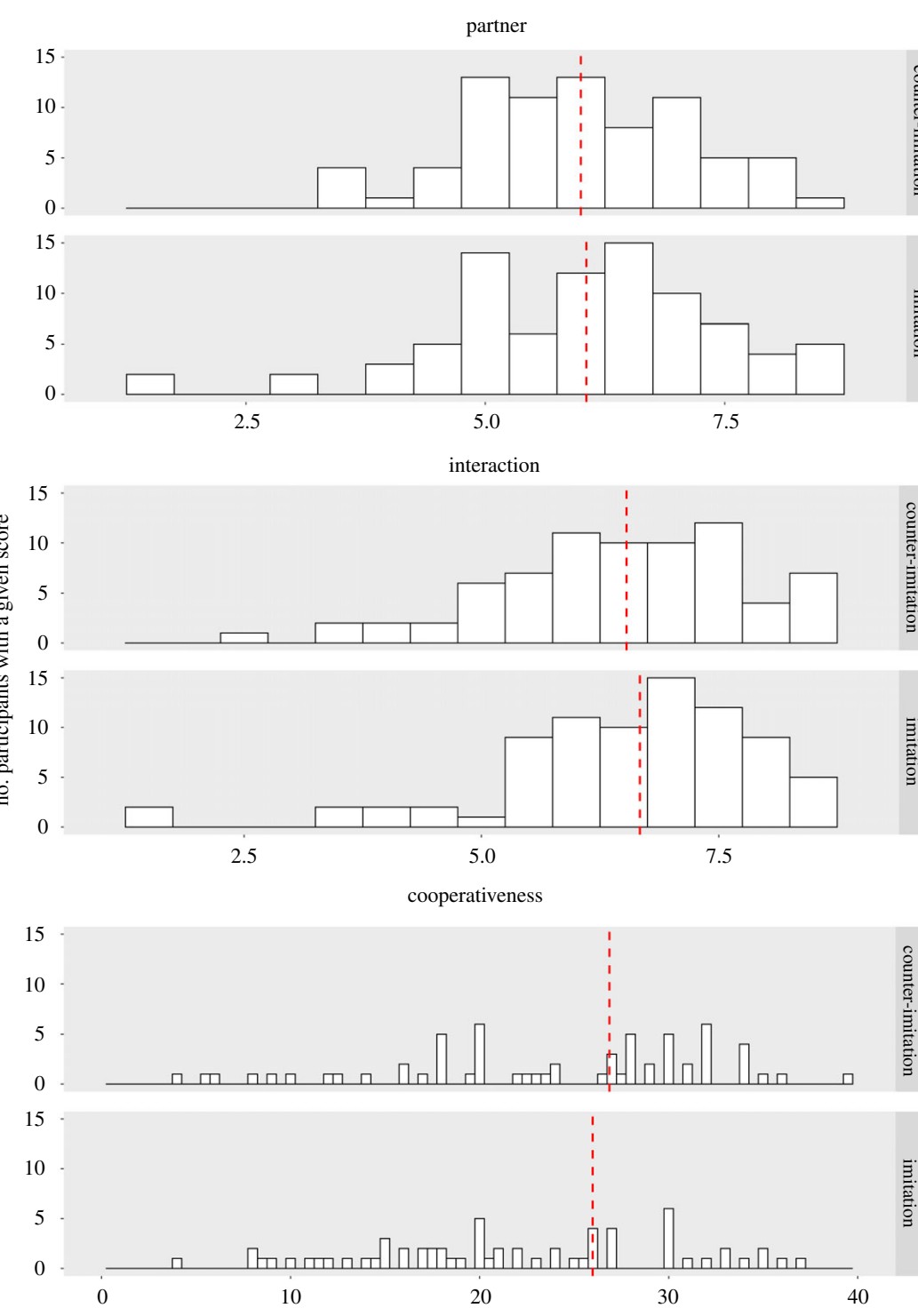

**Figure 1.** (Continued.)

tell the partner about themselves. Once participants submitted their introduction, a waiting screen appeared ('Waiting for partner's response', onscreen for 12 000–15 000 ms), followed by a text allegedly produced by the partner ('HI! I like dancing and swimming. From music I like pop, rnb, hip hop, grunge, etc. My favorite food is Indian food.My favorite place on Earth is Berlin.'; note the missing spaces and peculiarities in capitalization—these were introduced to support the cover story that the message was written by a human partner). Participants continued to the picture naming/ matching game whenever ready.

**Table 2.** Results from the linear regression models testing for the effect of condition (counter-imitation versus imitation), the effect of AQ and their interaction, on: participants' evaluation of the social interaction (interaction), their evaluation of the partner (partner) and their tendency to cooperate (cooperativeness).

| predictors | estimates | CI | p |
|---|---|---|---|
| interaction | | | |
| intercept | 6.56 | 6.25–6.88 | <.001 |
| AQ | 0.10 | −0.25–0.46 | .574 |
| condition | 0.19 | −0.24–0.63 | .378 |
| condition × AQ | −0.23 | −0.68–0.22 | .309 |
| observations | 147 | | |
| $R^2/R^2$ adjusted | 0.014/−0.007 | | |
| partner | | | |
| intercept | 6.04 | 5.75–6.33 | <.001 |
| AQ | 0.17 | −0.16–0.49 | .311 |
| condition | 0.07 | −0.33–0.47 | .738 |
| condition × AQ | −0.23 | −0.64–0.19 | .277 |
| observations | 147 | | |
| $R^2/R^2$ adjusted | 0.010/−0.011 | | |
| cooperativeness | | | |
| intercept | 26.78 | 24.37–29.19 | <.001 |
| AQ | −1.48 | −4.24–1.27 | .290 |
| condition | −0.98 | −4.35–2.40 | .568 |
| condition × AQ | 1.24 | −2.26–4.73 | .485 |
| observations | 147 | | |
| $R^2/R^2$ adjusted | 0.011/−0.010 | | |

Second, at the end of the experiment and after the debriefing questions, we added a questionnaire measuring the prevalence of autistic traits in the individual. We used the 10-item version of Autism-Spectrum Quotient (AQ-10) because it is short, has high validity [25] and covers communicative and attentional factors (both of which have been hypothesized to be relevant for lexical imitation; [24,26]). Items probed participants for behaviours and experiences indicative of autistic traits and used a four-step scale from 'definitely agree' to 'definitely disagree'. The study ended following this questionnaire, and participants were informed about the aims of the research and paid.

## 3.4. Results

Once again, we found no effects of being lexically imitated: there was no difference between the imitated and counter-imitated participants in terms of interaction evaluation ($U = 2493$, $p = 0.423$), partner evaluation ($U = 2605.5$, $p = 0.716$) or cooperativeness ($U = 2885$, $p = 0.472$; Mann–Whitney $U$-tests, one-tailed). See means in table 1 and score distributions in figure 1.

Further, none of these measures were affected by individual differences in the prevalence of autistic traits (participants' AQ score): for each measure, we conducted a linear regression model testing for the effect of condition (counter-imitation versus imitation), the effect of AQ and their interaction. The models found no statistically significant effects. See table 2 for model results.

## 4. General discussion

It has long been hypothesized that linguistic mimicry, much like the imitation of motor movements, can be used to improve interactions with others (e.g. [5,6]). In a pre-registered study involving lexical

imitation, we found no evidence to support this view: in Experiments 1–2, participants were assigned to a partner who either imitated or counter-imitated their lexical choices, and we measured how they evaluated the interaction with the partner and the partner themselves, and how willing they were to cooperate with the partner in a subsequent task; in Experiment 2, we also examined the prevalence of autistic traits in participants. We found that imitated participants did not reliably differ from counter-imitated participants on any of the three measures, and there was no modulatory effect of the prevalence of autistic traits on how participants responded to imitation/counter-imitation.

These data contrast with the positive findings reported in the literature. Some of the past studies implied that linguistic imitation may support positive interpersonal relations and pro-social behaviours, and even led to monetary gains for the imitator (e.g. [14]). Likewise, our previous work suggested that lexical imitation induces a positive evaluation of the interaction and encourages cooperation with the imitator [2]. In the present study, however, we found no evidence for such effects.

What could explain the absence of the hypothesized effects? One possibility relates to the context in which mimicry occurred in our study: it is plausible that the social benefits of linguistic imitation emerge only when the partner's behaviour is treated as a socially meaningful cue. This, in turn, depends on whether speakers interpret the exchange with the partner as a social interaction. Note that, depending on the context, the same behaviour may or may not be interpreted as informative of the partner or the interaction. To give an example, tapping one's foot can be taken as simply tapping to an imaginary rhythm or as a sign of being irritated with the current situation. What differentiates the two interpretations is social context—tapping the foot as a sign of irritation is likely only in the presence of other people (it would be rather surprising to see someone tapping with irritation while looking into the sea). Similarly, hearing another person say 'rubber' after we said 'eraser' might not prompt any concerns if we do not consider the exchange to be an actual social interaction.

Thus, in the current experiments, participants might not have considered the picture naming/matching task to be a social interaction and therefore did not interpret the partner's imitative/counter-imitative behaviour as socially meaningful. Recall that the experiments were implemented online. As a consequence, participants did not see or hear the partner with whom they were supposed to perform the task (nor did they come in contact with any of the experimenters). This distinguishes the current study from previous research where participants interacted face-to-face with the imitator (e.g. [11,12,14,16]), and from Lelonkiewicz [2] where participants were able to at least see their partner prior to the experiment (although they did not converse at that stage and later performed the same picture naming/matching task as here). To interpret an action as a social rather than an individual task, agents must form some minimal representations of their partner [27,28]. The fact that participants here completed the experiment in complete isolation might have discouraged them from creating mental representations for the partner and from interpreting the task as an interaction. As a result, they might not have processed the imitative/counter-imitative behaviour of the partner the way they would have if the partner had been physically present or if they had a more extensive interaction with the partner. Indeed, there is some evidence that alignment is sensitive to whether we interact with a stranger or a friend [29,30].

An alternative explanation of our null results of course relates to statistical power. However, as outlined in the Methods section, the present study had considerable statistical power, thus making it unlikely that the results constitute a false negative (each of the present experiments involved over 140 participants, which was sufficient to detect a medium-sized statistical effect with 80% power). A similar conclusion can be drawn from inspection of the descriptive patterns in the data, which suggests that the imitated and counter-imitated participants scored extremely similarly on all measures (note the condition means and 95% CI in table 1; see also figure 1).

In conclusion, our study contributed to the findings currently reported in the literature by focusing on the consequences of copying lexical choices. In two experiments, we found no evidence that this type of imitation leads to a more favourable evaluation of the imitator, a more favourable evaluation of the social interaction or greater cooperation. Beyond contributing by transparently reporting null results, our investigation points to a possible role of social context in the processing of linguistic mimicry. Future studies could examine the gains evoked by imitation in interactions with varying degrees of partner presence. Most importantly, we join the calls for more controlled experiments, including pre-registered experiments and replications, as the best way toward identifying the function of linguistic imitation (e.g. [17]).

Data accessibility. The data involved in this article can be accessed at: https://osf.io/z34m8/.

Authors' contributions. All authors contributed to the development of the study concept and design. Testing, data collection and analysis were performed by J.R.L. J.R.L. drafted the manuscript, and M.J.P. and H.P.B. provided critical revisions. All authors approved the final version of the manuscript for submission.

Competing interests. At the time of writing, M.J.P. was a Board Member of Royal Society Open Science, but had no involvement in the review or assessment of the paper.

Funding. This research was supported by Economic and Social Research Council grant no. ES/N013115/1 (awarded to H.P.B.) and research funds from Scuola Internazionale Superiore di Studi Avanzati (awarded to J.R.L.). Ethical approval (8-2021/1) was granted by Edinburgh University Psychology Research Ethics Committee.

Acknowledgements. We would like to thank Alisdair Tullo and Alessio Isaja for their help with developing the online paradigm. We thank Violetta Del Pinto for contributing to data collection of Experiment 1.

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
