## [Peer Review File · Royal Society Open Science]

Review History

RSOS-211107.R0 (Original submission)

Review form: Reviewer 1

Is the manuscript scientifically sound in its present form?

Yes

Are the interpretations and conclusions justified by the results?

Yes

Is the language acceptable?

Yes

Do you have any ethical concerns with this paper?

No

Have you any concerns about statistical analyses in this paper?

No

Recommendation?

Accept with minor revision (please list in comments)

Comments to the Author(s)

This is an elegant little paper with a clear goal. There are two studies, both of them establishing an online naming task that induces the appearance of "imitation" in an imagined task partner. The computer gives participants the appearance of imitation (e.g., using the same term for something). Experiment 1 is the basic task, and Experiment 2 has an "ice breaker" sort of setup. Neither showed evidence that imitation improves perception of the task partner or the interaction.

The authors do a good job integrating these null effects with theoretical discussion. The experiments are pre-registered, and their design carefully tailored to identify potential outcomes imitation. The fact that they obtain none is important theoretically -- as the authors discuss, prior work would predict a different pattern of results (including in their own work).

I do not have any major misgivings about this paper except for the obvious fact that the context of interaction/imitation is extremely limiting -- it is a scripted computer task run online, surely depriving participants with patterns of dynamism that may be expected in a more open-ended task with more co-presence. The authors of course address this and other limitations in the general discussion. I suspect this has quite a large effect, but of course we will only know this if researchers are prompted to conduct a similar study, in which the effects of imitation are measured and explored in on-site tasks where such modulating factors may be explored. That is a unique feature of this particular experiment: A surprisingly small number of studies explore the effects of imitation in a rigorously, experimentally controlled manner. That is another positive here.

Perhaps the only concern I have is that there is some other work on imitation that the authors may wish to consider. For example, Miles and others have experimentally manipulated perceived synchrony and explored social judgments as an effect of this (along with other intriguing manipulations from this group). I offer a sample citation below the authors may wish to consider.

In addition, there are several papers by Riordan and colleagues that explore alignment / mimicry in text-based communication, very relevant here. She and her collaborators have found that familiarity can have quite a large effect (e.g., strangers vs. friends as participants). This may lend some support to the issues raised in the general discussion.

On balance, I think this is a simple, elegant and pre-registered pair of studies revealing what may be considered theoretically important null effects. I leave my remarks above as general suggestions, but I endorse this work for publication in RSOS.

Miles, L. K., Nind, L. K., & Macrae, C. N. (2009). The rhythm of rapport: Interpersonal synchrony and social perception. *Journal of experimental social psychology*, 45(3), 585-589.

Riordan, M. A., Kreuz, R. J., & Olney, A. M. (2014). Alignment is a function of conversational dynamics. *Journal of Language and Social Psychology*, 33(5), 465-481.

Riordan, M. A., Markman, K. M., & Stewart, C. O. (2013). Communication accommodation in instant messaging: An examination of temporal convergence. *Journal of Language and Social Psychology*, 32(1), 84-95.

Review form: Reviewer 2

Is the manuscript scientifically sound in its present form?

Yes

Are the interpretations and conclusions justified by the results?

Yes

Is the language acceptable?

Yes

Do you have any ethical concerns with this paper?

No

Have you any concerns about statistical analyses in this paper?

No

Recommendation?

Accept with minor revision (please list in comments)

Comments to the Author(s)

This was an interesting and very well written paper, presenting a rigorous investigation of the influence of lexical imitation on social interaction. The questions being asked are timely, and the methods/analyses are replicable. I completely agree that the limited social interaction in the online naming task is likely to have interfered with imitation effects, and it will be interesting for future research to explore this possibility further. I only have some minor suggestions that the authors might want to work on before the paper is accepted.

1. The decision to assess individual differences in autistic traits in Experiment 2 would benefit from greater justification- both in terms of why these traits might relate to lexical imitation effects, but also why this particular measure was chosen (i.e. there are other measures that could have been used to assess sensitivity to social cues, whereas the AQ10 covers a broad range of autistic characteristics including local processing, cognitive flexibility, etc).
2. What happened on trials in which the participant made typos in their answers, which would then be transferred to the 'partner's' answer, or might result in an incorrect counter-imitation response being initiated? How often did this happen during the experiment?
3. It wasn't clear to me how the two post-interaction questionnaires were distinct from each other? Was it simply a presentation thing, i.e. the questions were split in two groups to hide the purpose and focus on partner?
4. The histograms should be moved to an appendix.
5. I recommend combining Tables 2-4 into a single Table with sub-headings for each measure. This would save space, make clearer which analyses refer to which DV, and enable easier comparison across the measures.
6. Page 6, paragraph 3 there is a word missing in the sentence, "...social benefits of linguistic imitation emerge only when [the] partner's behavior...".

Review form: Reviewer 3

Is the manuscript scientifically sound in its present form?

Yes

Are the interpretations and conclusions justified by the results?

Yes

Is the language acceptable?

Yes

Do you have any ethical concerns with this paper?

No

Have you any concerns about statistical analyses in this paper?

No

Recommendation?

Accept with minor revision (please list in comments)

Comments to the Author(s)

The manuscript describes findings from two experiments that addressed the question whether lexical imitation increases the perceived quality of a social interaction (partner), as well as fostering cooperation. The experiments served as an online replication of previous work by the authors. The authors did not find evidence for the hypothesized effect of being imitated.

The experiments address a relevant question and are well-executed. Thus, the manuscript provides a valuable contribution to the research field. Methods and results are motivated and presented in a clear and comprehensible way, and the authors provide a thoughtful interpretation of their findings, which they appropriately put into perspective by acknowledging potential limitations of their study. The manuscript is well-written.

I only have several minor comments.

1. In the abstract, the wording of the phrase “participants took turns” is somewhat misleading, given that participants were not truly interacting with other participants. It would be good to adapt the wording a bit, e.g. “participants took turns with a feigned partner” or something along those lines.
2. Page 4, first line: “People are prolific imitators – we copy the behaviour we see in others, from gestures to facial expressions to different aspects of language”. It would be good provide a reference here.
3. Page 6, lines 49-50: “In two laboratory experiments, participants took turns naming pictures and matching pictures to a name provided by a partner” It would be helpful to provide the reader with a bit more information about the set-up of the cited experiments: at this point in the manuscript, it is unclear for the reader whether the interaction between participants was in-person or online, or a mix of both - and hence, difficult to see the differences/similarities between these experiments and the current replication.
4. Page 20, second paragraph: I find the wording of this paragraph a little problematic, since it seems to be suggested that the present findings are incongruent with “the” previous literature and previous work by the authors. Yet, the picture that has emerged from the review of the

literature in the introduction is that evidence for an effect is mixed at best: previous work by others and by the authors themselves has yielded evidence for an effect in some cases, and a lack of any effects in others. So, I think it would be fair to rephrase the tone of this paragraph somewhat, acknowledging that current evidence for an effect of lexical imitation is mixed. Indeed, the authors should not disregard the possibility that the failure to detect the hypothesized effects could, in fact, be due to the lack of a true effect.

Decision letter (RSOS-211107.R0)

Dear Dr Lelonkiewicz

On behalf of the Editors, we are pleased to inform you that your Manuscript RSOS-211107 "Does it pay to imitate? No evidence for social gains from lexical imitation" has been accepted for publication in Royal Society Open Science subject to minor revision in accordance with the referees' reports. Please find the referees' comments along with any feedback from the Editors below my signature.

Please submit your revised manuscript and required files (see below) no later than 7 days from today's (ie 06-Oct-2021) date. Note: the ScholarOne system will 'lock' if submission of the revision is attempted 7 or more days after the deadline. If you do not think you will be able to meet this deadline please contact the editorial office immediately.

on behalf of Dr Giorgia Silani (Associate Editor) and Essi Viding (Subject Editor)
openscience@royalsociety.org

Associate Editor Comments to Author (Dr Giorgia Silani):

Comments to the Author:

The paper has been now reviewed by three experts in the field. They all agree that it addresses a relevant question and provides a valuable contribution to the research field. Furthermore they acknowledge that the studies are well-executed and the findings properly discussed. Nevertheless, they raised few points that the authors should address before consideration for publication.

Reviewer comments to Author:

Reviewer: 1

Comments to the Author(s)

This is an elegant little paper with a clear goal. There are two studies, both of them establishing an online naming task that induces the appearance of "imitation" in an imagined task partner. The computer gives participants the appearance of imitation (e.g., using the same term for something). Experiment 1 is the basic task, and Experiment 2 has an "ice breaker" sort of setup. Neither showed evidence that imitation improves perception of the task partner or the interaction.

The authors do a good job integrating these null effects with theoretical discussion. The experiments are pre-registered, and their design carefully tailored to identify potential outcomes imitation. The fact that they obtain none is important theoretically -- as the authors discuss, prior work would predict a different pattern of results (including in their own work).

I do not have any major misgivings about this paper except for the obvious fact that the context of interaction/imitation is extremely limiting -- it is a scripted computer task run online, surely depriving participants with patterns of dynamism that may be expected in a more open-ended task with more co-presence. The authors of course address this and other limitations in the general discussion. I suspect this has quite a large effect, but of course we will only know this if researchers are prompted to conduct a similar study, in which the effects of imitation are measured and explored in on-site tasks where such modulating factors may be explored. That is a unique feature of this particular experiment: A surprisingly small number of studies explore the effects of imitation in a rigorously, experimentally controlled manner. That is another positive here.

Perhaps the only concern I have is that there is some other work on imitation that the authors may wish to consider. For example, Miles and others have experimentally manipulated perceived synchrony and explored social judgments as an effect of this (along with other intriguing manipulations from this group). I offer a sample citation below the authors may wish to consider.

In addition, there are several papers by Riordan and colleagues that explore alignment / mimicry in text-based communication, very relevant here. She and her collaborators have found that familiarity can have quite a large effect (e.g., strangers vs. friends as participants). This may lend some support to the issues raised in the general discussion.

On balance, I think this is a simple, elegant and pre-registered pair of studies revealing what may be considered theoretically important null effects. I leave my remarks above as general suggestions, but I endorse this work for publication in RSOS.

Miles, L. K., Nind, L. K., & Macrae, C. N. (2009). The rhythm of rapport: Interpersonal synchrony and social perception. *Journal of experimental social psychology*, 45(3), 585-589.

Riordan, M. A., Kreuz, R. J., & Olney, A. M. (2014). Alignment is a function of conversational dynamics. *Journal of Language and Social Psychology*, 33(5), 465-481.

Riordan, M. A., Markman, K. M., & Stewart, C. O. (2013). Communication accommodation in instant messaging: An examination of temporal convergence. *Journal of Language and Social Psychology*, 32(1), 84-95.

Reviewer: 2

Comments to the Author(s)

This was an interesting and very well written paper, presenting a rigorous investigation of the influence of lexical imitation on social interaction. The questions being asked are timely, and the methods/analyses are replicable. I completely agree that the limited social interaction in the online naming task is likely to have interfered with imitation effects, and it will be interesting for future research to explore this possibility further. I only have some minor suggestions that the authors might want to work on before the paper is accepted.

1. The decision to assess individual differences in autistic traits in Experiment 2 would benefit from greater justification- both in terms of why these traits might relate to lexical imitation effects, but also why this particular measure was chosen (i.e. there are other measures that could have been used to assess sensitivity to social cues, whereas the AQ10 covers a broad range of autistic characteristics including local processing, cognitive flexibility, etc).
2. What happened on trials in which the participant made typos in their answers, which would then be transferred to the 'partner's' answer, or might result in an incorrect counter-imitation response being initiated? How often did this happen during the experiment?
3. It wasn't clear to me how the two post-interaction questionnaires were distinct from each other? Was it simply a presentation thing, i.e. the questions were split in two groups to hide the purpose and focus on partner?
4. The histograms should be moved to an appendix.
5. I recommend combining Tables 2-4 into a single Table with sub-headings for each measure. This would save space, make clearer which analyses refer to which DV, and enable easier comparison across the measures.
6. Page 6, paragraph 3 there is a word missing in the sentence, "...social benefits of linguistic imitation emerge only when [the] partner's behavior...".

Reviewer: 3

Comments to the Author(s)

The manuscript describes findings from two experiments that addressed the question whether lexical imitation increases the perceived quality of a social interaction (partner), as well as fostering cooperation. The experiments served as an online replication of previous work by the authors. The authors did not find evidence for the hypothesized effect of being imitated.

The experiments address a relevant question and are well-executed. Thus, the manuscript provides a valuable contribution to the research field. Methods and results are motivated and presented in a clear and comprehensible way, and the authors provide a thoughtful interpretation of their findings, which they appropriately put into perspective by acknowledging potential limitations of their study. The manuscript is well-written.

I only have several minor comments.

1. In the abstract, the wording of the phrase “participants took turns” is somewhat misleading, given that participants were not truly interacting with other participants. It would be good to adapt the wording a bit, e.g. “participants took turns with a feigned partner” or something along those lines.
2. Page 4, first line: “People are prolific imitators – we copy the behaviour we see in others, from gestures to facial expressions to different aspects of language”. It would be good provide a reference here.
3. Page 6, lines 49-50: “In two laboratory experiments, participants took turns naming pictures and matching pictures to a name provided by a partner” It would be helpful to provide the reader with a bit more information about the set-up of the cited experiments: at this point in the manuscript, it is unclear for the reader whether the interaction between participants was in-person or online, or a mix of both - and hence, difficult to see the differences/similarities between these experiments and the current replication.
4. Page 20, second paragraph: I find the wording of this paragraph a little problematic, since it seems to be suggested that the present findings are incongruent with “the” previous literature and previous work by the authors. Yet, the picture that has emerged from the review of the literature in the introduction is that evidence for an effect is mixed at best: previous work by others and by the authors themselves has yielded evidence for an effect in some cases, and a lack of any effects in others. So, I think it would be fair to rephrase the tone of this paragraph somewhat, acknowledging that current evidence for an effect of lexical imitation is mixed. Indeed, the authors should not disregard the possibility that the failure to detect the hypothesized effects could, in fact, be due to the lack of a true effect.

===PREPARING YOUR MANUSCRIPT===

If you have been asked to revise the written English in your submission as a condition of publication, you must do so, and you are expected to provide evidence that you have received language editing support. The journal would prefer that you use a professional language editing service and provide a certificate of editing, but a signed letter from a colleague who is a native

speaker of English is acceptable. Note the journal has arranged a number of discounts for authors using professional language editing services (<https://royalsociety.org/journals/authors/benefits/language-editing/>).

===PREPARING YOUR REVISION IN SCHOLARONE===

-- If you have uploaded ESM files, please ensure you follow the guidance at <https://royalsociety.org/journals/authors/author-guidelines/#supplementary-material> to include a suitable title and informative caption. An example of appropriate titling and captioning

may be found at https://figshare.com/articles/Table_S2_from_Is_there_a_trade-off_between_peak_performance_and_performance_breadth_across_temperatures_for_aerobic_sc_ope_in_teleost_fishes_/3843624.

Author's Response to Decision Letter for (RSOS-211107.R0)

See Appendix A.

Decision letter (RSOS-211107.R1)

Dear Dr Lelonkiewicz,

I am pleased to inform you that your manuscript entitled "Does it pay to imitate? No evidence for social gains from lexical imitation" is now accepted for publication in Royal Society Open Science.

Please see the Royal Society Publishing guidance on how you may share your accepted author manuscript at <https://royalsociety.org/journals/ethics-policies/media-embargo/>. After publication, some additional ways to effectively promote your article can also be found here

<https://royalsociety.org/blog/2020/07/promoting-your-latest-paper-and-tracking-your-results/>.

on behalf of Dr Giorgia Silani (Associate Editor) and Essi Viding (Subject Editor)
openscience@royalsociety.org

Appendix A

Dear Dr Silani,

We are very pleased to know that the Reviewers found our work important and interesting. We attach a new version of the manuscript, revised according to their suggestions. Below, in our response to the Reviewers, we report the changes to the manuscript. We would like to thank you again for considering our work.

With best regards,

Jaroslav R. Lelonkiewicz, Martin Pickering, Holly Branigan

Reviewer: 1

This is an elegant little paper with a clear goal. There are two studies, both of them establishing an online naming task that induces the appearance of "imitation" in an imagined task partner. The computer gives participants the appearance of imitation (e.g., using the same term for something). Experiment 1 is the basic task, and Experiment 2 has an "ice breaker" sort of setup. Neither showed evidence that imitation improves perception of the task partner or the interaction.

The authors do a good job integrating these null effects with theoretical discussion. The experiments are pre-registered, and their design carefully tailored to identify potential outcomes imitation. The fact that they obtain none is important theoretically -- as the authors discuss, prior work would predict a different pattern of results (including in their own work).

I do not have any major misgivings about this paper except for the obvious fact that the context of interaction/imitation is extremely limiting -- it is a scripted computer task run online, surely depriving participants with patterns of dynamism that may be expected in a more open-ended task with more co-presence. The authors of course address this and other limitations in the general discussion. I suspect this has quite a large effect, but of course we will only know this if researchers are prompted to conduct a similar study, in which the effects of imitation are measured and explored in on-site tasks where such modulating factors may be explored. That is a unique feature of this particular experiment: A surprisingly small number of studies explore the effects of imitation in a rigorously, experimentally controlled manner. That is another positive here.

Perhaps the only concern I have is that there is some other work on imitation that the authors may wish to consider. For example, Miles and others have experimentally manipulated perceived synchrony and explored social judgments as an effect of this (along with other intriguing manipulations from this group). I offer a sample citation below the authors may wish to consider.

In addition, there are several papers by Riordan and colleagues that explore alignment / mimicry in text-based communication, very relevant here. She and her collaborators have found that familiarity can have quite a large effect (e.g., strangers vs. friends as participants). This may lend some support to the issues raised in the general discussion.

Miles, L. K., Nind, L. K., & Macrae, C. N. (2009). The rhythm of rapport: Interpersonal synchrony and social perception. *Journal of experimental social psychology*, 45(3), 585-589.

Riordan, M. A., Kreuz, R. J., & Olney, A. M. (2014). Alignment is a function of conversational dynamics. *Journal of Language and Social Psychology*, 33(5), 465-481.

Riordan, M. A., Markman, K. M., & Stewart, C. O. (2013). Communication accommodation in instant messaging: An examination of temporal convergence. Journal of Language and Social Psychology, 32(1), 84-95.

We thank the Reviewer for a positive assessment of our work! We agree that the papers by Riordan et al. are very relevant for our discussion of the role of social context for alignment – we now refer to them on p. 22. However, we decided to not refer to the paper by Miles et al. as it is concerned with imitation of motor movements not language, and so is too distant from the focus of our study.

On balance, I think this is a simple, elegant and pre-registered pair of studies revealing what may be considered theoretically important null effects. I leave my remarks above as general suggestions, but I endorse this work for publication in RSOS.

Thank you!

Reviewer: 2

This was an interesting and very well written paper, presenting a rigorous investigation of the influence of lexical imitation on social interaction. The questions being asked are timely, and the methods/analyses are replicable. I completely agree that the limited social interaction in the online naming task is likely to have interfered with imitation effects, and it will be interesting for future research to explore this possibility further. I only have some minor suggestions that the authors might want to work on before the paper is accepted.

Thank you very much for these constructive comments.

1. The decision to assess individual differences in autistic traits in Experiment 2 would benefit from greater justification- both in terms of why these traits might relate to lexical imitation effects, but also why this particular measure was chosen (i.e. there are other measures that could have been used to assess sensitivity to social cues, whereas the AQ10 covers a broad range of autistic characteristics including local processing, cognitive flexibility, etc).

We clarified the reasons for measuring autistic traits (in the description of Experiment 2 on p. 17) and elaborated on the choice of the AQ-10 questionnaire (methods, pp 18-19).

2. What happened on trials in which the participant made typos in their answers, which would then be transferred to the 'partner's' answer, or might result in an incorrect counter-imitation response being initiated? How often did this happen during the experiment?

The experimental script was to an extent resistant to mistakes: it ignored any non-letter characters added to a response and was indifferent to lower/upper case. In addition, participants were instructed to avoid spelling mistakes. But as you suggest, in principle it was possible that certain spelling mistakes could have interfered with our imitation/counter-imitation manipulation.

Specifically, in the imitation condition, participant responses were copied verbatim, which could include spelling mistakes (e.g., participant types 'casette' and the program shows 'casette' as the partner's imitative response); in the counter-imitation condition, such mistakes could impair the ability of the experiment to recognize the word used by the participant, leading it to effectively show the same, but correctly spelled word, rather than an alternative name for the picture (e.g., participant types 'casette' and the program shows 'cassette', rather than 'tape').

Importantly, however, spelling mistakes occurred at a very low rate (1.29% of all experimental trials in E1 and 1.35% in E2), and so it is unlikely that they interfered with our manipulation to a noticeable extent.

For the information of the interested readers, we added a footnote discussing this issue (footnote 1).

3. It wasn't clear to me how the two post-interaction questionnaires were distinct from each other? Was it simply a presentation thing, i.e. the questions were split in two groups to hide the purpose and focus on partner?

See the description of methods on pp 12-13. The two questionnaires measuring the evaluation of the interaction and the partner were presented separately because they were in fact two different tools and relied on somewhat different response scales. That said, our previous work suggested that the data from these questionnaires can be merged without affecting the pattern of results (Lelonkiewicz, 2017).

4. The histograms should be moved to an appendix.

Thank you for this suggestion, but we would prefer to keep the histograms in the main text. We believe that these figures help the reader to quickly grasp the structure of the data and understand that, indeed, there were no statistical patterns (which to some might be surprising).

5. I recommend combining Tables 2-4 into a single Table with sub-headings for each measure. This would save space, make clearer which analyses refer to which DV, and enable easier comparison across the measures.

Thanks for this useful suggestion. The tables have been merged into Table 2.

6. Page 6, paragraph 3 there is a word missing in the sentence, "...social benefits of linguistic imitation emerge only when [the] partner's behavior..."

Well-spotted, thank you! We corrected this (p. 21).

Reviewer: 3

The manuscript describes findings from two experiments that addressed the question whether lexical imitation increases the perceived quality of a social interaction (partner), as well as fostering cooperation. The experiments served as an online replication of previous work by the authors. The authors did not find evidence for the hypothesized effect of being imitated.

The experiments address a relevant question and are well-executed. Thus, the manuscript provides a valuable contribution to the research field. Methods and results are motivated and presented in a clear and comprehensible way, and the authors provide a thoughtful interpretation of their findings, which they appropriately put into perspective by acknowledging potential limitations of their study. The manuscript is well-written.

I only have several minor comments.

Thank you for a positive appraisal of our work!

1. In the abstract, the wording of the phrase "participants took turns" is somewhat misleading, given that participants were not truly interacting with other participants. It would be good to

adapt the wording a bit, e.g. “participants took turns with a feigned partner” or something along those lines.

Point taken – we have revised the abstract accordingly.

2. Page 4, first line: “People are prolific imitators – we copy the behaviour we see in others, from gestures to facial expressions to different aspects of language”. It would be good provide a reference here.

We now added citations to two recent literature reviews (p. 3).

3. Page 6, lines 49-50: “In two laboratory experiments, participants took turns naming pictures and matching pictures to a name provided by a partner” It would be helpful to provide the reader with a bit more information about the set-up of the cited experiments: at this point in the manuscript, it is unclear for the reader whether the interaction between participants was in-person or online, or a mix of both - and hence, difficult to see the differences/similarities between these experiments and the current replication.

We clarified that, before in our laboratory experiments, participants carried out the task using computers, while seated in different rooms (p. 5).

4. Page 20, second paragraph: I find the wording of this paragraph a little problematic, since it seems to be suggested that the present findings are incongruent with “the” previous literature and previous work by the authors. Yet, the picture that has emerged from the review of the literature in the introduction is that evidence for an effect is mixed at best: previous work by others and by the authors themselves has yielded evidence for an effect in some cases, and a lack of any effects in others. So, I think it would be fair to rephrase the tone of this paragraph somewhat, acknowledging that current evidence for an effect of lexical imitation is mixed. Indeed, the authors should not disregard the possibility that the failure to detect the hypothesized effects could, in fact, be due to the lack of a true effect.

Thank you – we revised the wording of this paragraph to be more cautious and better reflect the complex character of the current findings (p. 21).

Lelonkiewicz, J. R. (2017). *Cognitive mechanisms and social consequences of imitation* (Doctoral dissertation, University of Edinburgh, Edinburgh, United Kingdom). Retrieved from <https://era.ed.ac.uk/handle/1842/23490>